# Outbreak of acute gastroenteritis associated with drinking water in rural Kazakhstan: A matched case-control study

**Madina Orysbayeva**[1,2‡], **Balaussa Zhuman**[1,3‡], **Dinara Turegeldiyeva**[1,4], **Roberta Horth**[5*], **Bakhytkul Zhakipbayeva**[5], **Daniel Singer**[5], **Manar Smagul**[4], **Dilyara Nabirova**[5]

**1** Central Asia Field Epidemiology Training Program, Almaty, Kazakhstan, **2** Ministry of Health, Committee of Sanitary and Epidemiological Control, Nur-Sultan, Kazakhstan, **3** Scientific and Practical Center for Sanitary and Epidemiological Expertise and Monitoring, Almaty, Kazakhstan, **4** M. Aikimbayev Kazakh Scientific Center of Quarantine and Zoonotic Diseases, Almaty, Kazakhstan, **5** U.S. Centers for Disease Control and Prevention, Central Asia Office, Almaty, Kazakhstan

‡ MO and BZ contributed equally and share first authorship to this work.
* hxw5@cdc.gov

**Data Availability Statement:** A limited anonymized dataset is available through a public repository. A limited anonymized dataset is available through the

## Abstract

We conducted an outbreak investigation from June 3 to 15th in a rural village in northern Kazakhstan, after surveillance showed an increase in gastroenteritis. Cases were residents who presented for medical treatment for diarrhea, fever (>37.5 ˚C), vomiting, or weakness from May 14 to June 15, 2021. Controls were residents matched by age ±2 years at a ratio of two controls for every case. Cases and controls were interviewed using structured questionnaires. We abstracted clinical data from medical records. We mapped cases and assessed risk for disease using conditional multivariable logistic regression. We identified 154 cases of acute gastroenteritis (attack rate of ~26 per 1,000 inhabitants). Symptoms were diarrhea, fever, vomiting, weakness, and decreased appetite. Among cases that participated (n = 107), 74% reported having drank unboiled tap water vs 18% of controls (n = 219). This was the only risk factor associated with disease (adjusted odds ratio: 18; 95% CI 9–35). Drinking water from a dispenser or carbonated drinks was protective. The city has two water supply networks; cases were clustered (107 cases in 79 households) in one. The investigation found that monitoring of quality and safety of water according to national regulations had not been conducted since 2018. No fatalities occurred, and no associated cases were reported after our investigation. Results suggest that untreated tap water was the probable source of the outbreak. The water supply had been cleaned and disinfected twice by the facility 2 days before our investigation began. Recommendations were made for regular monitoring of water supply facilities with rapid public notification when issues are detected to reduce likelihood of future drinking water associated outbreaks.

figshare repository (https://figshare.com/s/
6b22e1b58de35280aceb).

**Funding:** This activity was funded by the U.S.
Centers for Disease and Control and Prevention
Cooperative Agreement #GH20-2108 and by the
Kazakhstan Ministry of Health. The findings and
conclusions in this report are those of the author(s)
and do not necessarily represent the official
position of the U.S. Centers for Disease Control
and Prevention. The funders had no role in study
design, data collection and analysis, decision to
publish, or preparation of the manuscript.

**Competing interests:** The authors have declared
that no competing interests exist.

## Background

Despite global progress in decreasing the burden of enteric diseases, enteric pathogens remain
a major cause of morbidity and mortality globally, in the Central Asia region, including
Kazakhstan [1]. Globally over 1.6 million intestinal infection cases are registered annually, and
over 11 thousand cases were registered in Kazakhstan in 2019. North Kazakhstan accounted
for 384 cases in 2019. Inadequate water, sanitation, and hygiene infrastructure are the main
risk factors for enteric diseases in Central Asia [2].

From May 27 to June 1, 2021, district epidemiologists at the Kyzylzhar District Department
of Sanitary and Epidemiological Control noted an uptick in the number of cases of unspecified
gastroenteritis, registered in the village of 12,000 inhabitants in North Kazakhstan region,
Kazakhstan. A team from the Scientific and Practical Center for Sanitary and Epidemiological
Expertise and Monitoring and from the Field Epidemiology Training Program in Central Asia
Region were deployed to the village to conduct an outbreak investigation to describe epidemi-
ological and clinical characteristics, determine associated factors, and identify the source to
control the outbreak.

## Methodology

### Study design

The team conducted an outbreak investigation using a case-control study from June 3 to 15,
2021. Cases were defined as acutely ill residents of a rural village who had been hospitalized or
in outpatient treatment at any healthcare facility in the village from May 14 to June 15, 2021,
with signs and symptoms of diarrhea, fever (at least 37.5C), vomiting, and weakness. Controls
were residents of the village without these symptoms. Simple random sampling from the resi-
dent database, which includes demographic information on all people who resides in the
village, available from the Kyzylzhar District Department of Sanitary and Epidemiological
Control was used to select controls matched by age ±2 years at a ratio of 2 controls for every
case.

### Ethics statement

This investigation was reviewed and received approval as a non-human activity by U.S. Cen-
ters for Disease Control and Prevention under 45 CFR 46.102(*l*.2) and the Kazakhstan Minis-
try of Health Committee No. 24-03-21/2703 dated June 03, 2021. All participants gave written
informed consent. Parents or legal guardians gave consent for providing information about
their children under 18 years old. No children under 18 were interviewed, instead information
about children were collected from parents or legal guardians.

### Data collection

A list of cases was obtained from the Kyzylzhar District Department of Sanitary and Epidemio-
logical Control. To find additional cases, the team reviewed medical records for all persons
who became acutely ill and sought medical help from May 14 to June 15, 2021, with suspected
gastroenteritis infection. They reviewed bacterial culture laboratory data to find additional
cases among children hospitalized in the regional hospital. However, due to limited laboratory
capacity, most patients with gastroenteritis are treated syndromically.

Investigators used a structured questionnaire to assess demographic, epidemiological, clini-
cal, and laboratory characteristics. They developed a list of potential risks or exposures associ-
ated with gastroenteritis in the area. Risks included visiting rivers, lakes, swimming pools,
water parks, public catering places, trips outside the village, contact with a sick person, use of

non-boiled tap water, boiled tap water, dispenser water, wells, bottled water, and food products consumed during the last 14 days. Risk factors were evaluated for the 2 weeks prior to symptom onset in cases. Clinical and laboratory data were extracted from medical records.

## Data analysis

Patient data was entered and analyzed in EpiInfo 7.2.3.1. software (CDC, Atlanta). Univariate and bivariate analyses were performed to detect associations with disease. Statistically significant variables (p<0.01) were selected for conditional multivariate analysis. Multivariate logistic regression was performed to identify risk factors associated with gastroenteritis. Variable interactions and correlations were assessed. QGIS v.2.28 was used to detect clustering of cases.

## Laboratory testing

Fresh stool specimens were collected from 12 consenting cases, and water samples were collected from 2 consenting households. Virological and molecular genetic (PCR) analysis was carried out by the Scientific and Practical Center for Sanitary and Epidemiological Expertise and Monitoring Laboratory in Almaty.

## Results

### Descriptive characteristics

From 27 May to 13 June 2021, 154 persons suspected of gastroenteritis with clinical symptoms were identified as cases (Fig 1). This corresponds to an attack rate of 13 per 1,000 for the village and 26 per 1,000 in the old section of the village. Symptom onset peaked on May 27, 2021, prior to which there was the usual background incidence of enteric infections in the village and district. Cases occurred across age groups with no distinguishable age pattern over time. Most cases were children; 44% were <6 years old (n = 68/154) and 6% were <1 year old. The attack rate was 27 per 1,000 for the entire village and 53 per 1,000 for the old section of the village.

### Case-control study results

Of 154 cases, 107 (70%) cases were included in the study and matched to 219 controls. The remaining were excluded because of refusal (n = 12) or being away from their residence when the survey was conducted (n = 35). No deaths were recorded. Disease was mild in 13 cases and moderate in 94. The most commonly reported signs and symptoms were diarrhea (89%), vomiting (86%), abdominal pain (85%), nausea (76%), and fever ≥37.5˚C (65%) (Table 1).

Cases and controls did not differ by age or sex (42% and 44% of cases and controls were female, respectively, p = 0.8). Drinking unboiled tap water (odds ratio (OR): 12.5, 95% confidence interval (CI): 7.0–22.8) and having pets in the courtyard (OR 1.9, 95% CI 1.2–3.2) were independently associated with disease (Table 2). Drinking water from a dispenser, carbonated drinks, or preboiled tap water, eating kebabs, fish, berries and canned foods were protective. Drinking water from a pump, attending mass events, contact with a sick person, and bathing in open reservoirs, pools, and fountains were found in isolated cases in both groups and had no significant association with cases.

Drinking unboiled tap water (adjusted odds ratio (AOR) 18.2, 95% CI 9.4–35.3) was the only significant risk factor in multivariable analysis. Among cases, 54 (51%) noted an unpleasant smell, taste, and color of water before getting sick. Cases were clustered in households along the "old" part of the village (Fig 2). Of 79 households with cases, the majority had a single

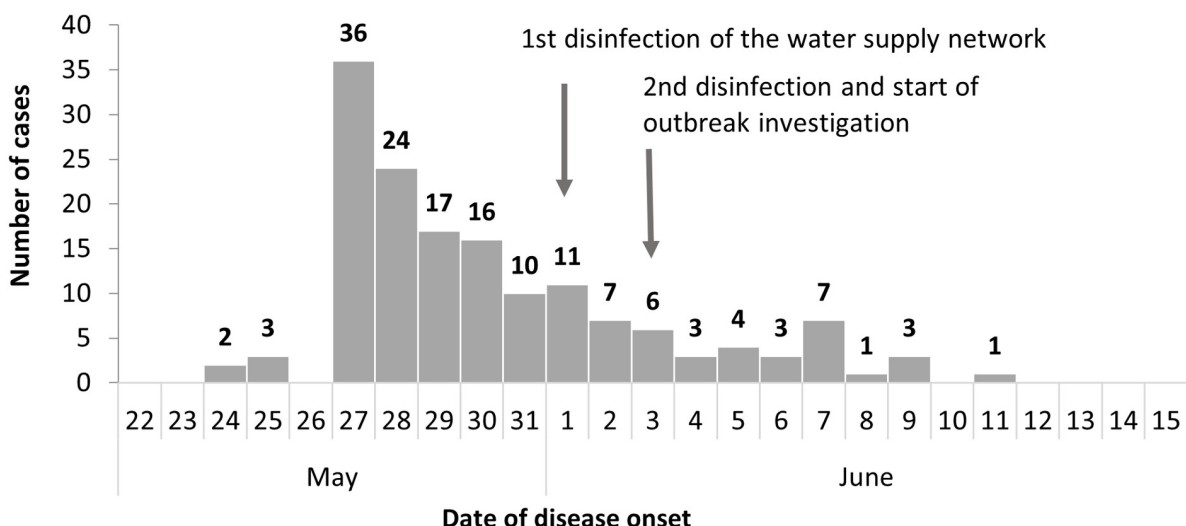

**Fig 1. Histogram of gastroenteritis in a rural village, Kazakhstan, May–June 2021 (N = 154).**

case (73%). But, 19 households had two case-patients each and two households had three case-patients each. In subanalysis, where only the first symptomatic case in each household was included, consumption of unboiled tap water remained significantly associated with disease (80% of cases vs. 18% of controls had drank unboiled tap water; OR: 17.4 [9.2–34.0]).

**Table 1. Characteristics of people with gastroenteritis in a rural village, Kazakhstan, May-2021.**

| Characteristics | Cases | Controls | p-value |
|---|---|---|---|
| | N (%) | N (%) | |
| **Total** | **107 (100)** | **219 (100)** | |
| Sex | | | 0.8 |
| Male | 45 (42) | 96 (44) | |
| Female | 62 (58) | 123 (56) | |
| Age (in years) | | | 0.9 |
| <1 | 7 (7) | 12 (5) | |
| 1–6 | 42 (39) | 80 (36) | |
| 7–17 | 23 (22) | 43 (20) | |
| 18–40 | 17 (16) | 39 (18) | |
| 41+ | 18 (17) | 45 (20) | |
| Symptoms | | | NA |
| Diarrhea | 95 (89) | 0 | |
| Vomiting | 92 (86) | 0 | |
| Abdominal pain | 91 (85) | 0 | |
| Nausea | 81 (76) | 0 | |
| Fever $\geq 37.5°C$ | 69 (65) | 0 | |
| Headaches | 29 (27) | 0 | |
| Muscle pain | 18 (17) | 0 | |
| Laryngitis | 13 (12) | 0 | |
| Cough | 11 (10) | 0 | |
| Rhinitis | 5 (5) | 0 | |
| Rash | 4 (4) | 0 | |

**Table 2. Factors associated with gastroenteritis in a rural village, Kazakhstan, May-June 2021.**

| Characteristics | Cases (n = 107) | Controls (n = 219) | OR | 95% CI | AOR | 95% CI |
|---|---|---|---|---|---|---|
| | n (%) | n (%) | | | | |
| Beverages consumed | | | | | | |
| Unboiled tap water | 79 (74) | 40 (18) | 12.5 | (7.0–22.8) | 18.2 | (9.4–35.3) |
| Carbonated drinks | 21(20) | 74 (34) | 0.5 | (0.3–0.8) | 0.4 | (0.2–0.8) |
| Boiled tap water | 74 (69) | 170 (78) | 0.6 | (0.4–1.1) | | |
| Filtered tap water | 18 (17) | 42 (19) | 0.9 | (0.4–1.6) | | |
| Bottled water | 48 (45) | 123 (56) | 0.6 | (0.4–1.0) | | |
| Water from a dispenser | 32 (30) | 100 (46) | 0.5 | (0.3–0.9) | 0.8 | (0.04–1.4) |
| Foods eaten | | | | | | |
| Kebab | 10 (9) | 47 (21) | 0.4 | (0.2–0.8) | 0.4 | (0.1–0.9) |
| Ice cream | 41 (38) | 105 (48) | 0.7 | (0.4–1.1) | | |
| Fruit | 87 (81) | 194 (89) | 0.6 | (0.3–1.1) | | |
| Fish | 13 (12) | 81 (37) | 0.2 | (0.1–0.5) | 0.8 | (0.5–1.5) |
| Berries | 9 (8) | 45 (21) | 0.4 | (0.1–0.8) | 0.4 | (0.2–1.1) |
| Canned foods | 6 (6) | 28 (13) | 0.4 | (0.1–1.0) | 0.4 | (0.1–1.5) |
| Animals contacted | | | | | | |
| Pets in the courtyard | 70 (65) | 109 (50) | 1.9 | (1.2–3.2) | 1.6 | (0.9–2.9) |
| People contacted | | | | | | |
| with a sick person | 16 (15) | 21 (10) | 1.7 | (0.9–3.4) | | |
| with a sick family member | 15 (14) | 19 (9) | 1.7 | (0.8–3.7) | | |
| Places visited | | | | | | |
| Visited a park | 8 (7) | 12 (5) | 1.4 | (0.5–3.8) | | |
| Visited a mall | 12 (11) | 23 (11) | 1.1 | (0.5–2.4) | | |

OR: Odds ratio, AOR: Adjusted odds ratio, CI: Confidence Interval.

## Laboratory testing results

Gastrointestinal pathogens were detected in 11 of the 12 cases that provided stool samples. Of these, all 11 cases had viral pathogens, 4 cases had 1 pathogen, and 7 had ≥2 pathogens. Viruses identified were norovirus, astrovirus, rotavirus, and enterovirus, including coxsackievirus (Table 3). No pathogens were detected in 2 tap water samples taken from two households, but samples were taken after chlorination of the water supply network (Fig 1).

## Water supply assessment results

The water supply in the village is centralized. The sewage systems are partially canalized in closed sewer pipes, sewage is discharged into a storage pond in the northwestern part of the village (at a distance of 3 km from the center of the village). Water is supplied by a nearby river. Water is treated (filtration, coagulation, disinfection with the use of liquid chlorine) at water treatment plants 22 km from the village and flows through main water pipelines to the village. The water supply then enters the city through different stations: the 'old' section of the village and the 'new' section of the village. The 'new' section of the village's water supply network was completely replaced, but the 'old' section of the village water supply has not been renovated for more than 30 years. In "old" section of the village water from the main water pipeline enters the pumping station into a water tower (1200 cubic meters), then into clean water tanks, and finally into the distribution network. The same company provides maintenance of the water supply network since 2017.

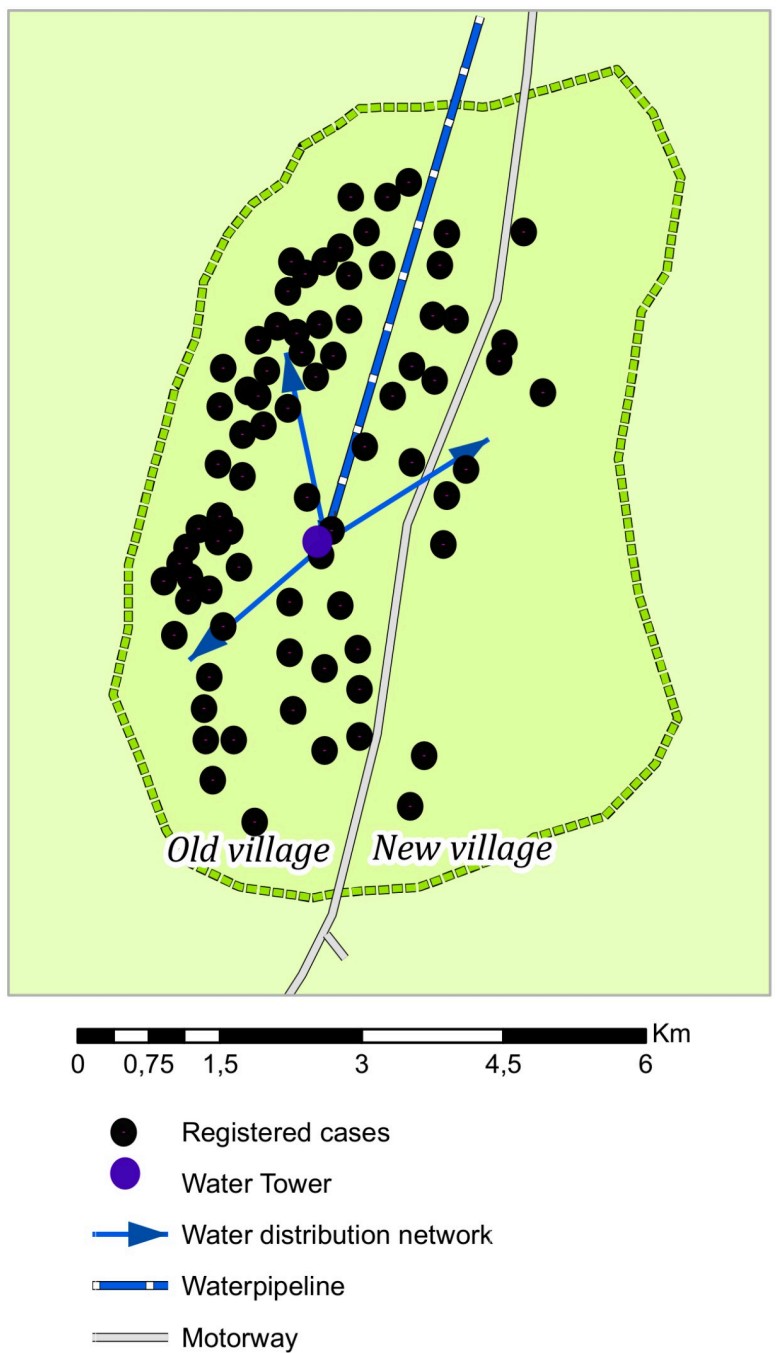

**Fig 2. Map of cases of gastroenteritis in a rural village, Kazakhstan, May–June 2021.**

Investigators found no records of water quality and safety control in the distribution network for the period 2018–2021. They also found no records for completed or missed procedures of water chlorination, testing of residual chlorine levels in the distribution system, and microbiological analysis of water in the 14 days prior to the first case at either the water supply company. They also did not find records of technical malfunctions and repairs among the centralized water supply networks.

**Table 3. Stool test results of cases with gastroenteritis (n = 12), Kazakhstan, May-June 2021.**

| Onset date | Virologic test results * | PCR test results** |
|---|---|---|
| 25.05 | negative | norovirus |
| 27.05 | negative | norovirus, astrovirus, rotavirus |
| 27.05 | negative | norovirus, astrovirus |
| 28.05 | negative | norovirus, astrovirus |
| 28.05 | negative | norovirus |
| 28.05 | negative | negative |
| 30.05 | negative | enterovirus, *Campylobacter* spp. |
| 30.05 | *Coxsackievirus B3* | rotavirus, enterovirus |
| 30.05 | negative | norovirus |
| 31.05 | negative | rotavirus |
| 01.06 | negative | enterovirus, astrovirus |
| 02.06 | *Coxsackievirus B5* | Enterovirus |

*Virological testing performed to detect enterovirus and coxsackievirus.

PCR testing was performed for the detection of enterovirus, norovirus genotype 2, astrovirus, rotavirus group A, adenovirus group F, *Listeria monocytogenes*, *Yersinia enterocolitica*, *Yersinia pseudotuberculosis*, *Shigella* spp., enteroinvasive *E. coli* (EIEC), *Salmonella* spp., *Campylobacter* spp.

District epidemiologists had tested water samples taken from 15 patient homes 4 days after the first case was reported. Their results found high levels of bacterial colony-forming units (CFUs) in water exceeding 5000 CFU per 100 mL. On 1 June 2021, the central water supply company cleaned and disinfected the water supply network in the village. They repeated cleaning and disinfection the following day, after testing showed noncompliance with the residual chlorine content in the water. On 3 June, the residual chlorine content was within normal limits.

## Discussion

From May 27 to June 13, 2021, an outbreak of gastroenteritis affected at least 26 per 1,000 inhabitants in the old section of village, Kyzylzhar district, North Kazakhstan region, Kazakhstan. The outbreak was likely due to contaminated drinking water. This conclusion is substantiated by the nearly 20-fold odds of having disease among persons who drank unboiled tap water and the decreased odds of having disease among persons who only drank from other sources (specifically water dispensers, bottled soda). Cases were also geographically concentrated in residences in the section of the rural village supplied by the same water supply network. Lastly, more than one viral pathogen was detected in half of the stool samples collected. This conclusion is further corroborated by findings from local epidemiologists of elevated total microbial contamination detected in tap water during the initial outbreak period and low levels of residual chlorine in tap water the day after disinfection of the water supply network.

The case-control study did not find any other common exposures. Disease was not associated with other water-related risks (such as recent visiting and bathing in a water park, fountains, an open pond) because the reports of these risks were equally rare among cases and controls. Similarly, there was no evidence linking cases to common events and activities, such as attending mass gatherings or dining. The mixed etiology of laboratory tests supports the hypothesis of drinking water contamination. The presence of multiple pathogens in the stool samples tested indicates that tap water could have been contaminated with feces from wastewater rather than from a single person. Wastewater contamination of municipal drinking

water supply systems has been a causal factor in other large outbreaks of acute intestinal infections in different countries. These outbreaks report mixed etiologies similar to those of this outbreak in a rural village, including norovirus, astrovirus, rotavirus, enterovirus, Coxsackievirus B3, and *Campylobacter* spp. [3–6]. An outbreak investigation of contaminated water supply in Greece similarly found that the majority of cases (6 of 11) had multiple pathogens detected in their stool [6].

Contamination of the water distribution system that has not been properly treated or maintained has been implicated in other outbreaks of water diseases in the region [7–10]. Lack of quality control of drinking water likely also contributed to the in a rural village outbreak [11]. Water supply systems in post-soviet countries have suffered from lack of maintenance and deterioration in rural areas. For example, a study in Uzbekistan found that risk for waterborne diseases in Uzbekistan is greater in rural areas [7]. The investigation found that the monitoring of the quality and safety of water according to national regulations had not been carried out since 2018, neither by the organization involved in the maintenance of the water supply facilities and the distribution network of the village nor by the supervisory. Although technical failures related to the water supply network were not officially documented or reported, several study participants reported having had concerns with the quality of their drinking water in the days preceding the outbreak.

Our investigation is subject to at least four limitations. First, few stool samples were collected from cases and no samples were collected from controls. No stool testing was performed at the local health facilities. The village is located in a low resource area and cases were empirically treated at local health facilities. Second, the outbreak team arrived after chlorination of the water system and collection of water from households do not reflect the water quality at the time the outbreak began. Third, the investigators were unable to perform a comprehensive inspection of the water supply system. Finally, the team was unable to carry out an active search for cases in the village, so the investigation likely underestimates the actual burden of gastroenteritis associated with this outbreak.

Our investigation demonstrates an independent association between the use of untreated tap water and disease and excludes the possibility of alternative explanations. When no etiological agent is detected in tap water, but several intestinal pathogens are isolated from more than half of the examined cases, and a clear epidemiological connection with the water factor of infection is established, we can be conclude that an outbreak is likely related to untreated water [12]. This is also consistent with the WHO approach, which defines an infection outbreak as water-borne if at least two people contract the same disease after drinking the same water, and epidemiological analysis identifies water as the source of infection [13]. Regular cleaning, washing and disinfection of water supply facilities in accordance with sanitary system regulations of Kazakhstan can reduce the likelihood of future drinking water associated outbreaks [14].

## Acknowledgments

**Disclaimer**: The findings and conclusions in this report are those of the author(s) and do not necessarily represent the official position of the U.S. Centers for Disease Control and Prevention.

## Author Contributions

**Conceptualization:** Madina Orysbayeva, Balaussa Zhuman, Dinara Turegeldiyeva, Manar Smagul, Dilyara Nabirova.

**Formal analysis:** Madina Orysbayeva, Balaussa Zhuman, Dinara Turegeldiyeva, Roberta Horth, Dilyara Nabirova.

**Funding acquisition:** Daniel Singer.

**Investigation:** Madina Orysbayeva, Balaussa Zhuman, Dinara Turegeldiyeva, Bakhytkul Zhakipbayeva, Manar Smagul, Dilyara Nabirova.

**Methodology:** Balaussa Zhuman, Dinara Turegeldiyeva, Bakhytkul Zhakipbayeva.

**Project administration:** Dilyara Nabirova.

**Supervision:** Bakhytkul Zhakipbayeva, Daniel Singer, Manar Smagul, Dilyara Nabirova.

**Validation:** Roberta Horth, Manar Smagul, Dilyara Nabirova.

**Writing – original draft:** Madina Orysbayeva, Balaussa Zhuman, Dinara Turegeldiyeva, Roberta Horth, Bakhytkul Zhakipbayeva, Daniel Singer, Manar Smagul, Dilyara Nabirova.

**Writing – review & editing:** Madina Orysbayeva, Balaussa Zhuman, Dinara Turegeldiyeva, Roberta Horth, Bakhytkul Zhakipbayeva, Daniel Singer, Manar Smagul, Dilyara Nabirova.

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
