## [Decision Letter · Decision Letter 0]

26 Sep 2022

PGPH-D-22-00997

Outbreak of acute gastroenteritis associated with drinking water in rural Kazakhstan: A matched case-control study

Dear Dr. Horth,

Thank you for submitting your manuscript to PLOS Global Public Health. After careful consideration, we feel that it has merit but does not fully meet PLOS Global Public Health’s publication criteria as it currently stands. Therefore, we invite you to submit a revised version of the manuscript that addresses the points raised during the review process.

We look forward to receiving your revised manuscript.

Kind regards,

Everton Falcão de Oliveira, Ph.D

Academic Editor

Journal Requirements:

1 Please review your reference list to ensure that it is complete and correct. If you have cited papers that have been retracted, please include the rationale for doing so in the manuscript text, or remove these references and replace them with relevant current references. Any changes to the reference list should be mentioned in the rebuttal letter that accompanies your revised manuscript. If you need to cite a retracted article, indicate the article’s retracted status in the References list and also include a citation and full reference for the retraction notice.

2. Please insert an Ethics Statement at the beginning of your Methods section, under a subheading 'Ethics Statement'. It must include:

a) The name(s) of the Institutional Review Board(s) or Ethics Committee(s)

b) The approval number(s), or a statement that approval was granted by the named board(s) 

c) (for human participants/donors) - A statement that formal consent was obtained (must state whether verbal/written) OR the reason consent was not obtained (e.g. anonymity). NOTE: If child participants, the statement must declare that formal consent was obtained from the parent/guardian.

3. Please amend your detailed Financial Disclosure statement. This is published with the article. It must therefore be completed in full sentences and contain the exact wording you wish to be published.

4. We noticed that you used “data not shown” in the manuscript. We do not allow these references, as the PLOS data access policy requires that all data be either published with the manuscript or made available in a publicly accessible database. Please amend the supplementary material to include the referenced data or remove the references.

5. Please provide separate figure files in .tif or .eps format only and remove any figures embedded in your manuscript file. Please also ensure that all files are under our size limit of 10MB.

6. In the online submission form, you indicated that "An anonymized limited data set can be made available upon request". All PLOS journals now require all data underlying the findings described in their manuscript to be freely available to other researchers, either 1. In a public repository, 2. Within the manuscript itself, or 3. Uploaded as supplementary information.

Additional Editor Comments (if provided):

Reviewers' comments:

Reviewer's Responses to Questions

**Comments to the Author**

1. Does this manuscript meet PLOS Global Public Health’s publication criteria? Is the manuscript technically sound, and do the data support the conclusions? The manuscript must describe methodologically and ethically rigorous research with conclusions that are appropriately drawn based on the data presented.

Reviewer #1: Yes

Reviewer #2: Yes

Reviewer #3: Yes

2. Has the statistical analysis been performed appropriately and rigorously?

Reviewer #1: Yes

Reviewer #2: No

Reviewer #3: Yes

3. Have the authors made all data underlying the findings in their manuscript fully available (please refer to the Data Availability Statement at the start of the manuscript PDF file)?

Reviewer #1: No

Reviewer #2: Yes

Reviewer #3: Yes

4. Is the manuscript presented in an intelligible fashion and written in standard English?

Reviewer #1: Yes

Reviewer #2: Yes

Reviewer #3: Yes

5. Review Comments to the Author

Reviewer #1: 1. Abstract

Clarify whether these suspected cases were included in the case group.

2. Metodology

I suggest characterizing the control group together with the case group in table 1 in study design;

I suggest trying to better detail the frequency of exposure of the two groups (case and control) mainly to the use of unboiled tap water in data collection;

I suggest adding the graphs that demonstrate the statistical analyzes in data analysis;

I suggest explaining why only 12 samples were collected of fresh stool specimens in laboratory testing.

3. Discussion

I suggest deepening the discussion by comparing the results obtained in this research with other studies already published on the subject.

Reviewer #2: Two points drew attention: The presentation of the data in greater detail and that the investigation is an activity, not being considered research, implying that it does not need the authorization of the ethics committee, only from the Ministry of Health.

Reviewer #3: I suggest that you make the following revisions:

1- In the methodology, better specify the exposure, graduating the degree of exposure (for example: did you use inadequate water once or is this routine consumption?), in order to make the group more homogeneous.

2- Increase the discussion with other articles in the literature, especially those that include data from nearby populations that consume water with similar treatment.

3- The results are well described, but I suggest including graphs to illustrate the results in a more didactic way.

6. PLOS authors have the option to publish the peer review history of their article (what does this mean?). If published, this will include your full peer review and any attached files.

**Do you want your identity to be public for this peer review?** For information about this choice, including consent withdrawal, please see our Privacy Policy.

Reviewer #1: No

Reviewer #2: No

Reviewer #3: No

---

## [Editor Report · Decision Letter 1]

8 Nov 2022

Outbreak of acute gastroenteritis associated with drinking water in rural Kazakhstan: A matched case-control study

PGPH-D-22-00997R1

Dear Dr Horth,

We are pleased to inform you that your manuscript 'Outbreak of acute gastroenteritis associated with drinking water in rural Kazakhstan: A matched case-control study' has been provisionally accepted for publication in PLOS Global Public Health.

Best regards,

Everton Falcão de Oliveira, Ph.D

Academic Editor